# Perceived Stress, Well-Being, and Academic Performance of University Students During the First COVID-19 Lockdown: A Study of Portuguese, Spanish, and Brazilian Students

**DOI:** 10.3390/healthcare13040371

**Published:** 2025-02-10

**Authors:** Alexandra Marques-Pinto, Luís Curral, Maria R. Costa, Francisca Quadros, Saúl Neves de Jesus, Isabel Maria Martínez, António Roazzi, Sofia Oliveira

**Affiliations:** 1Centro de Investigação em Ciência Psicológica, Faculdade de Psicologia, Universidade de Lisboa, 1649-013 Lisbon, Portugal; lcurral@psicologia.ulisboa.pt; 2Faculdade de Psicologia, Universidade de Lisboa, 1649-013 Lisbon, Portugal; maria.r.costa@cscm-lx.pt (M.R.C.); franciscavfcquadros@gmail.com (F.Q.); 3Faculdade de Ciências Humanas e Sociais, Universidade do Algarve, 8005-139 Faro, Portugal; snjesus@ualg.pt; 4Department of Work and Organizational Psychology, Universitat Jaume I, 12071 Castelló de la Plana, Spain; isabel.martinez@uji.es; 5Department of Psychology, Federal University of Pernambuco, Recife 50670-901, Brazil; antonio.roazzi@ufpe.br; 6Business Research Unit (BRU), ISCTE—Instituto Universitário de Lisboa, 1649-026 Lisbon, Portugal; sofia.oliveira@iscte-iul.pt

**Keywords:** academic engagement, academic outcomes, college students, mental health, perceived demands, SARS-CoV-2 pandemic

## Abstract

**Background/Objectives**: The COVID-19 pandemic profoundly affected university students globally, exacerbating their already existing academic stress. This study investigates how the first COVID-19 lockdown (March–July 2020) differently impacted Portuguese, Spanish, and Brazilian university students’ perceived academic stress, personal well-being, academic engagement, and performance. **Methods**: An online survey collected responses from 1081 university students (78.17% female; Mage = 25.43 years, SD = 9.27). Qualitative data on academic stressors were analyzed using content analysis. Cross-country differences were assessed through chi-square analyses and ANOVAs. Hypotheses were tested with a mediation path analysis. **Results**: Emotional distress emerged as the most prevalent stressor (54%). The results evidence differences in how students from the three countries experienced their academic life during the first COVID-19 lockdown. Personal and academic well-being mediated stress’ effects on performance. **Conclusions**: These findings underscore the need for context-tailored interventions and proactive measures to support students’ academic engagement in challenging contexts, informing educators and policymakers alike.

## 1. Introduction

The COVID-19 pandemic outbreak, caused by the coronavirus SARS-CoV-2, led to a global health crisis, triggering significant global challenges in early 2020. Its high transmission rate and severe health implications led governments and health organizations worldwide to take urgent measures to curb its spread. In March 2020, the virus spread rapidly across the globe [1], prompting governments to introduce general lockdowns as a key mitigation strategy. Thus, most face-to-face activities, including those in higher education institutions, were suspended. In Europe, higher education institutions closed in Spain on 12th March [2], and, in Portugal, the government ordered the closure of universities from 16th March [3]. Shortly after, in South America, Brazil also closed universities across the country [4]. Amid these closures and lockdown measures, university students were confronted with an unexpected transition to an emergency period of online classes [5], among other unforeseen challenges in their everyday life, with potential negative outcomes for their mental health and well-being [6]. Recent studies on COVID-19’s impact on college students reveal significant lifestyle changes and increased mental health risks, although the full extent of the pandemic’s impact is still under study [7]. Moreover, different experiences have been described across countries, thus calling for comparative research [7]. For instance, during the first wave of COVID-19, anxiety levels were higher among students in the Southern Hemisphere, particularly in Brazil and Oceania, possibly because the academic year had only recently begun [6]. In Portugal, due to the transition to online learning, students were among those who perceived having a higher workload, had lower levels of participation and interest in classes, and displayed increased concern regarding their final evaluation [6,8]. In Spain, perhaps because this was one of the countries in Europe most affected by the first wave of COVID-19, most students presented moderate to severe psychological impacts because of the lockdown [9].

Despite the growing number of studies on the impacts of the first COVID-19 lockdown on university students (depicted as one of the most affected groups [10]), comprehensive literature in this field remains limited. Most of the studies are overly descriptive, with small samples from single academic fields and/or institutions [8], and focus primarily on students’ stress sources and mental health [5,6]. The impacts on other spheres of students’ lives, such as their academic performance, remain unclear. While some studies report decreased academic performance post-transition to online learning [11], others find no change [12,13] or even an improvement [14,15]. 

According to the Transactional Model of Stress [16] and the Job Demands and Resources Model (JD-R) [17], stress arises from a perceived imbalance between demands and the resources available to meet them, impacting emotional (e.g., well-being) and behavioral (e.g., performance) outcomes. As a result of the COVID-19 pandemic, the demands experienced by university students increased, which, based on these models, may directly affect the experience of stress and, consequently, students’ personal and academic well-being and performance. However, inconsistent results are found in the literature, thus reinforcing the need for additional comparative research to better understand the global impact of the first COVID-19 lockdown on college students [7]. Hence, the present study aims to understand differences in perceived academic stress factors, personal and academic well-being (engagement), and academic performance during the first COVID-19 lockdown of Portuguese, Spanish, and Brazilian university students. Additionally, it explores the role of perceived stress, personal well-being, and academic engagement in the explanation of academic performance during that time.

### 1.1. Impacts of the COVID-19 Pandemic on University Students: Perceived Stress and Well-Being

The literature indicates that both previous lockdowns and COVID-19 have adverse and lasting effects on the general population’s well-being and mental health, including increased depressive symptoms, anxiety, insomnia, frustration, boredom, irritability, stress levels, emotional exhaustion, and post-traumatic stress [18,19]. Worldwide research on the impacts of COVID-19 on specific groups suggests that female, full-time workers, and undergraduate students’ have been particularly affected in terms of well-being and mental health [6,10,20,21,22].

University students, acknowledged as a vulnerable group in regard to mental health, face challenges due to the developmental tasks of emerging adulthood [23] and stressors linked with transitioning to higher education [24]. The initial COVID-19 lockdown significantly affected university students’ lives, increasing uncertainty [6,25] and exacerbating mental health challenges within this already vulnerable demographic [22,26].

A global study encompassing 62 countries revealed that COVID-19 introduced new stressors for college students in the academic (e.g., adapting to online learning, changes in communication and support, new assessment methods, managing time and workload), social (e.g., social isolation, canceled plans, uncertainty about lockdown duration), economic (e.g., job insecurity, future education and career concerns), and emotional (e.g., fear, anxiety, boredom) domains [6,7,8]. These findings suggest that university students’ experiences with stress were heavily influenced by the COVID-19 pandemic on a worldwide scale. To further understand the specific academic stressors experienced by university students across different countries, the following research question was established:

**Q1**:
*What were the most frequent academic stress sources perceived by Portuguese, Spanish, and Brazilian university students during the first COVID-19 lockdown? Were there cross-country differences?*


According to the Transactional Model of Stress, stress arises from a cognitive appraisal of an imbalance between life demands and the available resources to respond to those demands [16]. Coping responses play a crucial role in successful adaptation and are considered to be consistent predictors of well-being, particularly among university students [16,24,27]. University students, lacking the coping resources to manage the unforeseen demands of the first COVID-19 lockdown, faced a higher risk of prolonged stress and poorer well-being, with previous studies confirming increased stress symptoms such as anxiety and depression, emotional exhaustion, concentration difficulties, and sleep impairment [6,7,8,10,20,26,28]. However, studies remain scarce and non-consensual. Some research suggests decreased well-being [7,22,26,28,29], while others report unchanged well-being levels [30]. These discrepancies likely stem from differences in COVID-19’s manifestation and progression across the studied countries, along with their respective responses. Therefore, to test the relationship between perceived academic stress intensity and well-being among university students during the first COVID-19 lockdown, the following hypothesis was tested:

**Hypothesis** **1.**
*Perceived academic stress intensity will be negatively associated with well-being.*


### 1.2. Impacts of Perceived Stress on University Students’ Academic Performance: The Mediating Role of Personal and Academic Well-Being

Although the COVID-19 lockdown had some positive effects, like improved self-care and autonomy through online learning, the negative impacts are generally seen to outweigh the positives. The positive effects are influenced by factors like institutional support and individual coping skills, such as self-regulated learning and social-emotional competencies [20]. Various studies have consistently highlighted the significant role of individual coping responses alongside perceived stress in the well-being and academic performance of college students [31,32,33]. However, findings on the impact of the pandemic on academic performance remain inconclusive [13]. To grasp the factors behind these inconsistent findings, researchers are exploring socioeconomic (e.g., income levels; [11]), contextual (e.g., teachers’ expectations, perceived support; [20,34]), and individual variables (e.g., attitude towards online learning, self-efficacy; [12,13]) as potential moderators of the pandemic’s impact on academic performance. However, further research is needed to deepen our understanding and mitigate the pandemic’s adverse effects on university students’ academic performance [7]. Therefore, to investigate the cross-country impacts of the first COVID-19 lockdown on university students, the following research question was established:

**Q2**:
*Did the first COVID-19 lockdown impact university students’ perceived academic stress intensity, well-being, engagement, and academic performance? Were there cross-country differences?*


Studies suggest that work context factors can impact students’ well-being in higher education. For example, one study [35] linked the perception of obstacles/facilitators to students’ burnout and engagement, while another [36] found associations between peer social support, working conditions, role clarity, and subjective well-being. Previous studies employing the JD-R [17] found that excessive academic demands coupled with inadequate resources increase the risk of prolonged stress among students, while adequate resources contribute to increased academic engagement [24,35]. In fact, academic engagement is considered a key indicator of university students’ well-being, since it is viewed as a fulfilling and positive work-related state of mind, characterized by a sense of vigor (energy, resilience, and persistence), dedication (commitment, enthusiasm, and purpose), and absorption (full concentration and immersion in work) [33]. It has also been linked to academic performance [31,33].

Therefore, in a context in which emergency online learning has defied students’ self-regulation, autonomy, self-efficacy, and resilience [20,37], academic engagement may be a relevant variable to help understand and further explain the COVID-19 pandemic’s impacts on students’ academic performance. Some studies have explored the effects of the COVID-19 outbreak and emergency online learning on university students’ engagement [38,39,40] and corroborate the role that students’ personal well-being and academic engagement may play in academic performance. Research has found that, during the COVID-19 outbreak, students’ personal well-being was positively associated with engagement [38] and that engagement was negatively associated with students’ views of emergency online experiences as unpleasant [37]. A study with Romanian students found decreased dedication and vigor during emergency online learning amid increased stress and reduced well-being, despite higher absorption [40]. Canadian students also reported a significant drop in engagement, coupled with lower perceptions of academic success [39]. Despite these preliminary results, to the best of our knowledge, the potentially protective role of students’ engagement in the relationship between academic stress and performance during the first COVID-19 lockdown is yet to be studied. In view of the above, to test the role of perceived stress, personal well-being, and academic engagement in the explanation of the academic performance of university students during the first COVID-19 lockdown, the following hypotheses were established regarding our second research aim:

**Hypothesis** **2.**
*Personal well-being will be positively associated with academic engagement.*


**Hypothesis** **3.**
*Academic engagement will be positively associated with academic performance.*


**Hypothesis** **4.**
*Personal well-being and academic engagement will mediate the relationship between perceived academic stress intensity and academic performance, and the indirect effect will be lower than the direct effect.*


The proposed model of this study, shown in Figure 1, conceptualizes university students’ personal well-being and engagement as mediators between their perceived academic stress and performance.

## 2. Materials and Methods

### 2.1. Participants and Procedure

Our sample comprised college students from Portuguese, Spanish, and Brazilian public universities. These countries were selected due to their varying impacts from the pandemic, despite certain similarities and geographical/cultural proximities [6]. We chose two countries with similar university systems and academic calendars, but differing impacts from the first wave of COVID-19 (e.g., Spain was severely affected compared to Portugal), along with a third country with a distinct academic calendar (e.g., Brazil’s academic year starts in March, unlike Portugal and Spain’s September start), to explore potential differences in students’ academic experiences during the initial COVID-19 lockdown. Within these countries, the sample was selected by convenience due to previous working relationships between the institutions.

The data were collected using an online survey between October 2020 and January 2021. Students were invited via email, sent by their respective university departments to the mailing list, and were asked to voluntarily complete an online questionnaire. The research team had no prior relationship with the participants, and no compensation was provided. Participants were informed about the research’s purpose at the start of the questionnaire and were asked for their consent to participate. Upon giving consent, participants completed the survey, which lasted an average of 20 min.

Participants completed surveys in their native language (Spanish or Portuguese) using validated versions of the measurements. They were instructed to refer to the period of the first lockdown (from March to July 2020). To enhance data validity, only complete responses were included [41]; text entry boxes were used for collecting sociodemographic data to identify random responses, spam, or autofill software usage [42], and a statement promoting honesty was included in the survey introduction to mitigate social desirability bias [43].

We used a convenience sample, acknowledging both its advantages and limitations. Convenience sampling is quick, straightforward, and cost-effective, as participants are selected based on availability and willingness, reducing time and financial burdens. It is particularly useful for exploratory research, providing initial insights or testing hypotheses in the early stages of a study. However, convenience sampling may lack generalizability due to the non-random selection of participants, potentially leading to selection bias and limited representativeness. Nonetheless, we collected data from several different universities to enhance diversity and minimize localized biases.

### 2.2. Measures

#### 2.2.1. Perceived Academic Stress

Students were asked to describe the three academic stressors they had most frequently experienced during the first COVID-19 lockdown. Additionally, overall academic stress intensity was measured with a single item by asking students, “To what extent do you consider that your academic activity, in general, was stress-generating over those months?” This item was rated on a 5-point Likert scale, ranging from 1 = not generating stress at all to 5 = extremely stress-generating.

#### 2.2.2. Personal Well-Being

We used the Mental Health Continuum–Short Form (MHC-SF) [44,45] to measure personal well-being through 14 items focusing on feelings of emotional, psychological, and social well-being. Answers were given on a 6-point Likert scale, ranging from 1 = never to 6 = every day.

#### 2.2.3. Academic Engagement

Academic engagement was measured with the 9-item short version of the Utrecht Work Engagement Scale for students (UWES-S) [33,46], which measures feelings of vigor, dedication, and absorption. Students were requested to report, on a 7-point Likert scale that ranged from 1 = never to 7 = every day, how often they had experienced those feelings.

#### 2.2.4. Academic Performance

Academic performance was measured by asking students to compare the average marks they obtained in the second semester course units of 2019/2020 with the average of those obtained in the first semester. Answers were given on a 3-point scale (lower, the same, or higher).

### 2.3. Data Analysis

Data analyses were performed using SPSS Statistics 24. Regarding perceived academic stress sources, a mixed deductive/inductive thematic content analysis was performed following Bardin’s guidelines [47]. Taking the JD-R [7] as a framework, a deductive analysis was initially conducted in which two overarching Demands and Resources categories were considered a priori. To deepen the analysis, an inductive analysis of the specific demands and (lack of) resources emerging from the students’ responses was then conducted, and they were added to the category system (e.g., demands for distance learning). A frequency analysis of each subcategory was subsequently performed. Since each participant could list up to three stressors, each subcategory was assigned a respective frequency of response, ranging from 0 (never mentioned) to 3 (mentioned three times). Chi-square analyses were performed between each subcategory according to nationality, and contingency tables were analyzed. To ensure the validity and reliability of the analysis, two coders initially categorized all responses following exclusivity, homogeneity, pertinence, objectivity, and productivity assumptions [47]. Then, a third independent coder categorized 100% of the responses into the category system. Divergences were discussed until an agreement percentage of 100% was attained. Descriptive statistics and ANOVAs were employed to examine cross-country variations in overall academic stress intensity, personal well-being, academic engagement, and academic performance. Pearson correlations were also computed.

Lastly, for our hypotheses testing, a Confirmatory Factor Analysis (CFA) was performed with Mplus 8 to ensure that the established multi-item scales were distinct from each other [48]. The hypothesized three-factor model (i.e., perceived academic stress intensity, personal well-being, and academic engagement) fit the data significantly better (χ^2^ (207) = 2145.51, *p* < 0.001) than both the baseline (Δχ^2^ (231) = 18,410.86, *p* < 0.001) and the one-factor (Δχ^2^ (211) = 17,503.49, *p* < 0.001) models. The standardized parameter estimates (factor loadings) of the best fitting three-factor model were all significant (*p* < 0.01) and ranged from 0.48 to 0.88. The measurement model demonstrated an acceptable fit: a normed comparative fit index (CFI) of 0.90, a standardized root-mean-square residual (SRMR) of 0.04, and a root-mean-square error of approximation (RMSEA) of 0.08. Overall, the indices demonstrated an acceptable fit [49]. Coefficient alphas of the multiple-item measurements were greater than the generally accepted threshold of 0.70 [50].

Our hypotheses were tested using manifest (observed) variable path analysis with PROCESS v4.0 macro for SPSS [51]. This methodology was adopted as it allows for the simultaneous modeling of individual and multiple mediation paths. In this model, there is no requirement for a total effect between X and Y to be present when testing for a mediation, as the focus of a mediation test is on the indirect effect of X on Y through the mediators [51,52]. Also, PROCESS made it possible to compare the indirect effects through various paths, including each mediator separately and a serial mediation through two sequential mediators. All of the paths were included in our models to avoid bias in the estimate of the serial indirect effect predicted in Hypothesis 4 [51]. Three demographic variables were included in the analysis as control variables: age (in years), sex (female = 1; male = 0), and nationality (by creating three dummy variables; Brazil = 0; Portugal = 1; Spain = 1). These variables were chosen since, following the prior literature, they appear to impact the variables under study.

## 3. Results

### 3.1. Sample Characteristics

The sample consisted of 1081 university students (78.17% female; *M* = 25.43 years, *SD* = 9.27), comprising 534 students from Portugal, 371 students from Spain, and 176 students from Brazil. The sample included undergraduate students (65%), as well as master’s (28%) and PhD students (7%). The degree programs attended by the students spanned various areas, ranging from the Humanities and Social Sciences, Legal and International Studies, Economics and Management, Engineering and Technology, to Health Sciences. Our sample also included student workers (Portugal = 23.4%, Spain = 36.1%, and Brazil = 37.4%). Participants who did not respond to one of the study variables or failed to provide information on gender, age, and education level were excluded from the sample, resulting in a total of 73 students being removed.

### 3.2. Perceived Academic Stress Sources

A full description of the perceived academic stress sources identified by the students during the first COVID-19 lockdown is depicted in Appendix A.

Concerning the situations perceived as more stressful during the COVID-19 lockdown, demands related to experiencing emotional distress were the most frequently mentioned, with 54% of the participants considering them a major stressor. In addition, work overload (29%), (lack of) institutional support (29%), remote classes (27%), assessment (25%), and isolation (22%) were considered major sources of stress by the participants. Also, the lack of technological resources was deemed a major source of stress for 10% of the participants. The remaining situations were regarded as major sources of stress for a proportion of the participants, ranging from 2% to 4%.

#### Cross-Country Differences on Academic Stress Sources

In a comparison of the countries using Chi-square tests, significant differences between Portugal and Spain were identified regarding the perception of demands related to experiencing emotional distress (χ^2^ = 5.76, *p* < 0.05), academic assessment (χ^2^ = 3.76, *p* < 0.05), remote classes (χ^2^ = 4.61, *p* < 0.05), isolation (χ^2^ = 4.11, *p* < 0.05), and the management of family, professional, and academic interests (χ^2^ = 8.71, *p* < 0.01). A significant difference between Portugal and Brazil was also found in the perception of demands related to experiencing emotional distress (χ^2^ = 5.52, *p* < 0.05). The frequency and proportions of stress sources by country are depicted in Table 1. In the comparisons where at least one cell had less than five observations, we used Fisher’s exact test (e.g., finance, space, and research). No significant differences were found.

### 3.3. Academic Stress Intensity, Well-Being, Engagement, and Academic Performance

The descriptive statistics and correlations among the remaining variables for the total sample are presented in Table 2. As expected, academic performance was positively related to personal well-being (*r* = 0.14, *p* < 0.01), positively related to academic engagement (*r* = 0.21, *p* < 0.01), and negatively related to perceived academic stress intensity (*r* = −0.13, *p* < 0.01). In addition, academic engagement was positively related to personal well-being (*r* = −0.57, *p* < 0.01) and negatively related to perceived academic stress intensity (*r* = −0.33, *p* < 0.01). Finally, among the control variables, females were more likely to perceive higher academic stress than males (*r* = 0.11, *p* < 0.01). However, there were no significant differences between males and females in terms of personal well-being (*r* = −0.04, *n.s.*) or academic engagement (*r* = 0.04, *n.s.*). In addition, the older participants were more engaged with academic work (*r* = 0.16, *p* < 0.01), had higher personal well-being (*r* = 0.17, *p* < 0.01), and perceived less academic stress (*r* = −0.11, *p* < 0.01) than the younger participants.

#### Cross-Country Differences

To ascertain the differences between the three sub-samples, a one-way ANOVA was conducted on perceived academic stress intensity, well-being, engagement, and academic performance. The means and standard deviation of all the variables for each country can be found in Table 3. The ANOVA results show that the Portuguese participants (*M* = 4.13, *SD* = 1.59) had significantly higher levels of engagement than the Spanish participants (*M* = 3.67, *SD* = 1.40), who, in turn, had significantly lower levels of engagement than the Brazilians (*M* = 4.01, *SD* = 1.58) (*F* = 10.17, *p* < 0.001). Similarly, the Portuguese participants (*M* = 3.56, *SD* = 1.14) had significantly higher levels of well-being than the Spanish (*M* = 3.09, *SD* = 0.95) and Brazilian participants (*M* = 3.29, *SD* = 1.09), and the Spanish participants had lower levels of well-being than the Brazilian participants (*F* = 21.95, *p* < 0.001). As for academic performance, the Brazilian participants (*M* = 2.03, *SD* = 0.70) indicated lower performance than the Portuguese (*M* = 2.17, *SD* = 0.76) and Spanish participants (*M* = 2.22, *SD* = 0.74), with no difference between those two countries (*F* = 3.85, *p* < 0.022). Finally, no significant differences were found in perceived academic stress intensity among the three countries (*F* = 1.33, *p* < 0.266).

In summary, the descriptive results highlight the key sources of academic stress, with experiencing emotional distress being the most commonly reported stressor among the participants. Cross-country comparisons reveal significant differences in how stressors, such as emotional distress, academic assessment, remote classes, isolation, and balancing family, professional, and academic interests, were perceived between Portugal, Spain, and Brazil. Despite these findings, no significant differences were found in perceived academic stress intensity across the three countries. Academic performance was positively associated with well-being and engagement and negatively related with academic stress intensity. Engagement was also positively related to well-being and negatively to academic stress intensity. Cross-country variations were also found in well-being, academic engagement, and academic performance. Notably, Portuguese participants reported the highest levels of well-being, while Spanish students reported the lowest levels of both personal well-being and academic engagement, and Brazilian students indicated the lowest academic performance compared to students from the other countries.

### 3.4. Test of Proposed Conceptual Model: Mediation Analysis

The serial multiple mediation results are provided in Table 4. As recommended by [52], all possible direct effects were included in the analysis to estimate the serial indirect effect. In line with our first hypothesis (Table 4, model 1), perceived academic stress intensity was negatively related to personal well-being (*b* = −0.32, *p* < 0.01). Hypothesis 2 (Table 4, model 2) was also corroborated, as personal well-being was positively related to academic engagement (*b* = 0.71, *p* < 0.01). In model 2, perceived academic stress intensity was still significant (*b* = −0.28, *p* < 0.01). Likewise, Hypothesis 3 (Table 4, model 3) was supported, with academic engagement being positively associated with academic performance (*b* = 0.09, *p* < 0.01). In model 3, personal well-being was no longer significant, and although the effect of perceived academic stress on academic performance remained significant, it decreased in size. These findings provide initial support for a mediation effect.

To test Hypothesis 4, that the relationship between perceived academic stress intensity and academic performance would be mediated sequentially through personal well-being and academic engagement, the indirect effects (reported in Table 4) were examined. From a comprehensive perspective, and as recommended by [51], three indirect paths were estimated and tested simultaneously using bootstrap confidence intervals based on 5000 bootstrap samples and using the PROCESS 4.3 tool. The indirect effect of perceived academic stress intensity on academic performance through personal well-being was non-significant (point estimate = −0.01; 95% CI [−0.03, 0.01]). Nevertheless, the indirect effect of perceived academic stress intensity on academic performance through academic engagement was negative and significant (point estimate = −0.03; 95% CI [−0.04, −0.01]). Consequently, the serial indirect effect of perceived academic stress intensity on academic performance through personal well-being and academic engagement was negative and significant (point estimate = −0.02; 95% CI [−0.03, −0.01]), thus supporting Hypothesis 4. Overall, the results suggest an indirect effect of perceived academic stress on academic performance through personal well-being and academic engagement.

## 4. Discussion

This study sought to investigate the impacts of the initial COVID-19 lockdown on university students from different countries. It aimed to identify key academic stressors and explore the role of perceived academic stress intensity, personal well-being, and academic engagement in the explanation of their academic performance during that period. Additionally, it examined differences in these factors among Portuguese, Spanish, and Brazilian college students, aligning with the literature emphasizing the importance of understanding the pandemic’s impacts across different countries.

### 4.1. Q1: What Were the Most Frequent Academic Stress Sources Perceived by Portuguese, Spanish, and Brazilian University Students During the First COVID-19 Lockdown? Were There Cross-Country Differences?

Despite being prompted to identify sources of academic stress, our findings indicate that demands related to experiencing emotional distress emerge as a primary stressor, acknowledged by over half of the participants. This underscores the challenge of managing emotional distress as a key impact of the COVID-19 pandemic. This strain, expressed through increased symptoms of anxiety and depression previously linked to COVID-19 [18,19], reproduces a direct impact on the well-being and health of the populations, particularly younger individuals [6,10,21]. The university students also identified work overload, lack of institutional support, remote classes, assessments, and isolation as major sources of academic stress stemming from the first COVID-19 lockdown, aligning with the prior literature that highlights similar challenges faced by university students in their academic pursuits amid the pandemic [7].

As for the cross-country comparisons on perceived sources of academic stress stemming from COVID-19, our findings suggest that the Portuguese students perceived less demands related to experiencing emotional distress compared to the Spanish and Brazilian students. Spanish students also referred to managing family, professional, and academic interests as being more stressful compared to Portuguese students. In turn, the Portuguese university students reported more stressors associated with the transition to online learning (i.e., remote classes, academic assessment) than the Spanish students. These significant differences in the perception of demands between Portugal, Spain, and Brazil likely reflect cultural, educational, and socio-contextual factors. Cultural norms regarding emotional expression and mental health may influence how students report emotional distress, while variations in academic systems and remote learning approaches could explain differences in perceptions of academic assessment and online classes. Additionally, disparities in social distancing measures and support systems during the study period may have contributed to differing experiences of isolation and the challenge of balancing family, professional, and academic responsibilities, particularly between Portugal and Spain.

### 4.2. Q2: Did the First COVID-19 Lockdown Impact University Students’ Perceived Academic Stress Intensity, Well-Being, Engagement, and Academic Performance? Were There Cross-Country Differences?

Descriptive statistics also show that, overall, university students experienced high academic stress levels, particularly females and younger participants, which is consistent with prior scientific evidence regarding COVID-19 [10]. Moreover, older participants exhibited higher academic engagement and personal well-being compared to younger participants. This aligns with the developmental psychology literature [23,24], suggesting that younger students may still be in the process of developing emotional coping mechanisms and adapting to university compared to their older counterparts. However, previous research on age-related differences in engagement and well-being is inconsistent. While some studies suggest a general increase in engagement with age [53], others report minimal or no age differences in engagement [46,54]. Similarly, results on personal well-being differences across the lifespan are not curvilinear or consistent between components: emotional well-being seems to decrease during adolescence and increase during adulthood, but the same does not happen with other well-being components (e.g., psychological) [55].

Regarding personal well-being and academic engagement, the pandemic appears to have had a significant impact. Our data suggest that both were rated slightly below the midpoint in the scale. Relatively recent pre-pandemic research with college students has depicted Portuguese university students with moderately high levels of personal well-being [36] and academic engagement [31], with mean values above the mid-point of the scale for both variables. The same pattern was found regarding Spanish university students, who also presented moderately high levels of academic engagement [31] in a pre-pandemic study. Conversely, our findings show that academic performance tended to remain similar to pre-pandemic results, according to the students. Taken together, these results suggest that, even though students were apparently able to maintain their academic performance, the first COVID-19 lockdown impaired their well-being and academic engagement.

Furthermore, while no cross-country differences emerged in perceived academic stress intensity, Portuguese students presented the highest levels of personal well-being (compared to the Spanish and Brazilian students), and Spanish students reported the lowest levels of both personal well-being and academic engagement (compared to the Portuguese and Brazilian students). Nevertheless, the Brazilian students indicated lower academic performance than the Portuguese and Spanish students.

These findings align with the previous literature [6,8] and may arise from variations in the experiences of the first COVID-19 lockdown among these countries. As Spain was one of the countries in Europe most affected by the first wave of COVID-19 [9], Spanish students might have faced heightened disruption, emotional distress, and challenges in balancing life roles. In contrast, Portugal experienced the first lockdown with less social alarm, early government intervention, and fewer cases and deaths, which may have potentially contributed to making adaptation to new teaching methods the primary challenge for students [20], thereby explaining the greater impact on academic life management in Portugal. Another explanatory hypothesis could be that the Portuguese university system was less prepared for this abrupt transition than its Spanish counterpart.

Lastly, in keeping with previous studies [6], Brazilian students exhibited greater emotional distress than Portuguese students, alongside lower academic performance rates compared to Portuguese and Spanish students. These findings might relate to the fact that Brazil’s academic year was in its early stages, and they may also depict the slightly higher prevalence of student workers in the Brazilian sample. Certain jobs, particularly those more precarious ones that support working students’ studies, were significantly impacted by the COVID-19 pandemic, potentially influencing the emotional well-being and academic performance of Brazilian students. Additionally, the difference in emotional distress perception between Portugal and Brazil may be linked to socioeconomic disparities, family structures, and access to support systems, which vary significantly between these countries. Furthermore, cultural attitudes toward education and societal expectations may influence how academic demands are experienced and reported. These findings highlight the importance of considering the cultural, educational, and socioeconomic contexts when interpreting cross-country comparisons of student perceptions.

### 4.3. Test of Hypotheses: Personal Well-Being and Academic Engagement Will Mediate the Relationship Between Perceived Academic Stress Intensity and Academic Performance, and the Indirect Effect Will Be Lower Than the Direct Effect

Additionally, our findings offer preliminary support for the mediating role of personal well-being and academic engagement in the relationship between perceived stress and university students’ academic performance, indicating that higher levels of personal well-being and academic engagement are associated with reduced stress effects on academic performance. However, the indirect effect was only significant in the presence of academic engagement, suggesting that the relationship between academic stress and academic performance relies more on academic engagement than on personal well-being [37,38,39]. This result aligns with previous studies, including pre-pandemic ones [31,45] and others following the pandemic outbreak [39], suggesting a positive association between academic engagement and college students’ academic performance. One possible explanation for this result is that engagement, conceptualized in relation to students’ academic tasks and activities [33], is directly associated with students’ perception of resources to meet academic demands [24,32], whereas personal well-being, encompassing a broader evaluation of emotional experience, life satisfaction, and psychosocial functioning [44], may be less directly linked to academic performance.

### 4.4. Limitations

Our study is not without its limitations. It relied on a non-probabilistic and female-overrepresented sample, limiting generalizability. Data were collected exclusively online during the pandemic, which, despite using a validation protocol, warrants further exploration of these variables and their associations, resorting to different data collection methods and a probabilistic and gender-balanced sample. The cross-sectional design, based on self-report questionnaires requiring recall of past experiences, introduces potential bias and prevents analysis of changes over time. Future research should examine these findings across disciplines (e.g., Science, Engineering, the Humanities) and educational levels (e.g., undergraduate, master’s, PhD) using probabilistic sampling to address gender differences and longitudinal dynamics highlighted in the literature.

## 5. Study Impact

Despite its limitations, our study furthers the understanding of perceived academic stress, personal well-being, and academic engagement’s role in explaining university students’ academic performance during the first COVID-19 lockdown. It sheds light on how the pandemic’s initial wave affected university students across the three studied countries, contributing to the understanding that the pandemic’s impacts should be considered within sociocultural contexts. This underscores the need for culturally sensitive interventions, given the varied global needs and experiences, even among culturally similar and geographically proximate countries like Portugal and Spain. For instance, in Portugal, interventions should prioritize supporting transitions to online learning by providing students with enhanced technological resources and more flexible assessment methods. In Spain, addressing the heightened emotional distress and challenges related to balancing family, professional, and academic roles should be a priority, perhaps through family-inclusive policies or time management support. In Brazil, given the lower academic performance and emotional distress reported, strategies should focus on offering financial and psychological support to working students, particularly those in precarious jobs severely impacted by the pandemic.

Lastly, the present study sustains the importance of educational interventions targeting students’ academic success to explicitly address students’ academic engagement and not only their emotional distress regulation [56]. Thus, actionable recommendations must be designed to address the specific stressors faced in each country, leveraging their respective educational systems’ strengths while addressing their weaknesses. For example, Portugal could benefit from training faculty in digital pedagogy, Spain from developing robust mental health services within universities, and Brazil from improving access to academic resources and stabilizing support for working students. The findings highlight that, for students to maintain a successful relationship with their academic activity, it is not sufficient for them to feel good in their personal life—they must also be engaged in their academic life. Furthermore, COVID-19 lockdown stressors identified by students represent risk factors for their engagement. Thus, the response should not solely focus on developing emotion-related coping mechanisms, but should also aim to maintain and increase student engagement (with a focus on improving performance), potentially by improving assessment and lecture processes and institutional support. Future strategies should include innovative approaches, such as leveraging technology to enhance remote learning experiences, creating culturally adaptive teaching methods, and incorporating specific policies to support the diverse realities of students across these countries.

## Figures and Tables

**Figure 1 healthcare-13-00371-f001:**
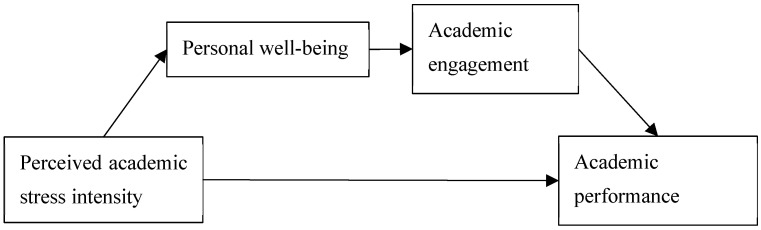
The proposed conceptual model.

**Table 1 healthcare-13-00371-t001:** Frequency and proportions of stress sources by country.

		WO	Distress	Assmnt	Remote	Isolation	Support	Tech	Finance	Space	Bibliographic	WLB	Research
Brazil	Sum	27	56	19	21	21	31	9	2	5	6	2	1
	%	30	62	21	23	23	34	10	2	5	7	2	1
Spain	Sum	78	153	56	62	47	68	25	12	6	6	19	5
	%	30	58	21	23	18	26	9	5	2	2	7	2
Portugal	Sum	89	151	89	98	77	98	31	13	16	16	7	10
	%	28	48	28	31	25	31	10	4	5	5	2	3
Total	Sum	194	360	164	181	145	197	65	27	27	28	28	16
	%	29	54	25	27	22	29	10	4	4	4	4	2

Note. Number of participants: Brazil = 91; Spain = 264; Portugal = 313; Total = 668. WO = demands related to work overload; Distress = demands related to experiencing emotional distress; Assmnt = demands related to assessment; Remote = demands related to remote classes; Isolation = demands related to isolation; Support = lack of resources related to institutional support; Tech = lack of technological resources; Finance = lack of financial resources; Space = lack of space resources; Bibliographic = lack of bibliographic resources; WLB = demands related to the management of family, professional, and academic interests; Research = demands related to practice/research impairment.

**Table 2 healthcare-13-00371-t002:** Descriptive statistics and correlations.

	Mean	SD	1	2	3	4	5
1. Academic performance	2.16	0.745					
2. Academic engagement	3.95	1.54	0.21 **	(0.93)			
3. Personal well-being	3.36	1.09	0.14 **	0.57 **	(0.92)		
4. Perceived academic stress intensity	3.73	0.97	−0.13 **	−0.33 **	-0.31 **	-	
5. Age	25.43	9.27	−0.05	0.16 **	0.17 **	−0.11 **	-
6. Gender	0.78	0.41	0.04	0.04	−0.04	0.11 *	−0.12 **

Note. *n* = 1081; * *p* < 0.05, ** *p* < 0.01. Reliabilities (Cronbach alphas) are reported in parentheses.

**Table 3 healthcare-13-00371-t003:** Differences between countries in stressors, engagement, well-being, and academic performance.

	Brazil	Portugal	Spain	Eta Squared	95% Confidence Lower Bound	95% Confidence Upper Bound
1. Academic performance	2.03 (0.70)	2.17 (0.76)	2.22 (0.74)	0.005	0.005	0.016
2. Academic engagement	4.01 (1.58)	4.13 * (1.59)	3.67 (1.40)	0.018	0.005	0.035
3. Personal well-being	3.29 (1.09)	3.56 (1.14)	3.09 (0.95)	0.047	0.025	0.072
4. Perceived academic stress intensity	3.71 (0.86)	3.69 (0.97)	3.80 * (1.03)	0.002	−0.002	0.010

Note. Portugal, *n* = 534; Spain, *n* = 371; Brazil, *n* = 176. Standard deviation in brackets. * significant differences at *p* < 0.05.

**Table 4 healthcare-13-00371-t004:** Mediation results.

	Model 1 Personal Well-Being b (SE)	Model 2 Academic Engagement b (SE)	Model 3 Academic Performance b (SE)
Control variables			
Age	0.02 (0.00) **	0.01 (0.00) **	−0.01 (0.00) **
Gender	0.01 (0.08)	0.31 (0.09) **	0.06 (0.05)
Portugal	0.34 (0.09) **	−0.07 (0.11)	0.09 (0.06)
Spain	−0.16 (0.09)	−0.19 (0.11)	0.22 (0.07) **
Independent variables			
Perceived academic stress intensity	−0.32 (0.03) **	−0.28 (0.04) **	−0.05 (0.02) *
Personal well-being	--	0.71 (0.04) **	0.03 (0.03)
Academic engagement	--	--	0.09 (0.02) **
R^2^	0.16 **	0.37 **	0.07 **
Bootstrap indirect effects:	Coefficient (SE)	LL 95% CI	UL 95% CI
PASI > Personal well-being > Academic performance	−0.01 (0.01)	−0.03	0.01
PASI > Academic engagement > Academic performance	−0.03 (0.01) *	−0.04	−0.01
PSI > Personal well-being > Work engagement > Academic performance	−0.02 (0.01) *	−0.03	−0.01

Note. *n* = 1081. CI = confidence interval; LL = lower limit; UL = upper limit; PASI = perceived academic stress intensity. * *p* < 0.05; ** *p* < 0.01.

## Data Availability

The raw data that support the findings of this study are available from the corresponding author, A.M.-P., upon reasonable request due to the informed consent agreement, which does not account for public sharing of the data.

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
