# Peer review of "Perceived Stress, Well-Being, and Academic Performance of University Students During the First COVID-19 Lockdown: A Study of Portuguese, Spanish, and Brazilian Students"

_healthcare, 2025, doi:10.3390/healthcare13040371_

Round 1

Reviewer 1 Report

Comments and Suggestions for Authors

I read with interests the paper titled "Perceived stress, well-being, and academic performance of university students during the first COVID-19 lockdown: A study of Portuguese, Spanish and Brazilian students"

I have some comments to the manuscript:

1. The four year gap between the data collection and its reporting is for me a major drawback of the paper. The novelty is low, and the changes that COVID created makes the conclusions not current anymore. 

2. First COVID lockdown was in February March 2020; however, data was collected several months after, meaning that the perception of the students could have been potentially changed by the gap between the occurence of the lockdown and the questionnaire application. 

3. Introduction is too long. Authors should state the hypothesis and briefly justify them; there is no need to formulate questions and answer with hypothesis that authors will try to answer afterwards. This could be useful for academic purposes, but not in a scientific paper. Keep it as simple as possible: formulate your problem, add a methodology, try to answer it. 

4. Again, no need two pages for methods. Keep it as shorten as possible. Line 220-226 are results. No need to explain in detail validated scales used.

5. Mean age is 25.43, which seems to me a high value for university students. In Portugal the most prevalent age is 18-22 (60.1%) (Source: Perfil do Aluno 2022/2023 - DGEEC); In Spain 54,6% have 18-21 years (Source:  Estadística de Estudiantes Universitarios (EEU) - 2022/2023);

Reviewer 2 Report

Comments and Suggestions for Authors

The topic is relevant for educators/ policymakers managing crises in higher education.

The novelty is somehow undermined. COVID-19 finished years ago, and numerous studies have already been published on this subject. The manuscript may need to specify new dimensions of the issue.

The generalizability is limited due to the convenience sample.

Clear objectives. Distinguishing between exploratory and hypothesis-testing goals would add clarity. The methodology is sound.

Results were well-presented but largely descriptive. Some findings are predictable (e.g., females reporting higher stress), diluting impact.

Recommendations are practical but lack innovative strategies. Greater emphasis on unique cross-country differences would enhance value.

The manuscript is well-organized, but there are grammatical issues and inconsistencies in citation style. Transitions between sections could be smoother.

Materials and Methods (Lines 208-316):

While it is mentioned university students from public universities in Portugal, Spain, and Brazil, it is unclear whether additional criteria (age range, academic field, enrollment status…) were applied to ensure the sample's relevance and homogeneity.

Include a detailed exclusion criteria to clarify how incomplete or invalid responses were handled and why participants from private universities, if excluded, were not considered.

Clarify justification for using convenience sampling. Acknowledge its limitations.

How biases like social desirability were statistically controlled.

Line 227-228: IRB approval redundant with Lines 558-560.

Results (Lines 318-406):

Provide richer cross-country comparisons with effect sizes.

Add a summary paragraph connecting descriptive and inferential results.

Table 2: The use of zero or omission of zero before a decimal must be consistent throughout the manuscript.

Discussion (Lines 407-505):

Strengthen implications with actionable recommendations tailored to each country.

Reduce speculative statements unsupported by data.

Line 441: format consistently for "Covid-19"

Limitations (Lines 506-520): integrate into discussion and limit to 2-3 critical issues.

The manuscript offers valuable insights but needs refinement in novelty framing, methodological justification, and implications.

Reviewer 3 Report

Comments and Suggestions for Authors

Thank you for the opportunity to review this manuscript. The manuscript “Perceived stress, well-being, and academic performance of university students during the first COVID-19 lockdown: A study of Portuguese, Spanish and Brazilian students” presents a well conducted study that investigates how the first COVID-19 lockdown impacted university students from Portugal, Spain and Brazil, their perceived academic stress, personal well-being, academic engagement, and performance. I have only a couple of comments suggested to be addressed before the next steps in publication.

1.     In the introduction section, I would suggest not to mix the pandemic outbreak and lockdown implementations. The start of the introduction section is misleading and mixing both together. I would suggest to kindly separate and introduce both of them separately/independently.

2.     According to study results, students were from different academic programs, i.e., undergraduate, master’s and doctorate students. I would suggest to analyze them separately as the stress levels for these disciplines are different. Also, they are affected differently, e.g., more affects for undergraduate students than doctorate students.

 Minor comments:

1.     I would suggest replacing “Academic performance”, “University students”, “Covid-19”, “Stress” and “Well-being” in key words with suitable similar words. As these words have already been used in the title of the manuscript.

Round 2

Reviewer 1 Report

Comments and Suggestions for Authors

No further comments to add. 

Reviewer 2 Report

Comments and Suggestions for Authors

The authors responded adequately all of my comments, I think the manuscript is improved now. I have no further comments.